# Coping Self-Efficacy and Its Relationship with Psychological Morbidity after Genetic Test Result Disclosure: Results from Cancer-Unaffected *BRCA1/2* Mutation Carriers

**DOI:** 10.3390/ijerph20031684

**Published:** 2023-01-17

**Authors:** Anna Isselhard, Zoe Lautz, Maren Töpper, Kerstin Rhiem, Rita Schmutzler, Frank Vitinius, Hannah Fischer, Birte Berger-Höger, Anke Steckelberg, Karolina Beifus, Juliane Köberlein-Neu, Stephanie Stock

**Affiliations:** 1Institute of Health Economics and Clinical Epidemiology, University Hospital Cologne, 50924 Cologne, Germany; 2Center for Hereditary Breast and Ovarian Cancer and Center for Integrated Oncology (CIO), Medical Faculty, University Hospital Cologne, 50924 Cologne, Germany; 3Department of Psychosomatics and Psychotherapy, Faculty of Medicine, University Hospital Cologne, 50924 Cologne, Germany; 4Institute for Public Health and Nursing Research, University of Bremen, 28359 Bremen, Germany; 5Institute for Health and Nursing Science, Faculty of Medicine Martin Luther University Halle-Wittenberg, 06112 Halle (Saale), Germany; 6Center for Health Economics and Health Services Research, Schumpeter School of Business and Economics, University of Wuppertal, 42119 Wuppertal, Germany

**Keywords:** *BRCA1*, *BRCA2*, self-efficacy, psychological burden, anxiety, breast cancer

## Abstract

Women who are found to carry a *BRCA1/2* pathogenic variant experience psychological distress due to an increased risk of breast and ovarian cancer. They may decide between different preventive options. In this secondary analysis of data collected alongside a larger randomized controlled trial, we are looking at 130 newly found *BRCA1/2* pathogenic variant carriers and how their coping self-efficacy immediately after genetic test result disclosure is related to their psychological burden and status of preventive decision making. Participants received the Coping Self-Efficacy Scale, the Hospital Anxiety and Depression Scale, the Impact of Event Scale, the Decisional Conflict Scale, and the Stage of Decision-Making Scale after positive genetic test result disclosure. We found that women with higher coping self-efficacy showed fewer symptoms of anxiety or depression and were less affected by receiving the genetic test result in terms of post-traumatic stress. However, coping self-efficacy had no relationship with any decision-related criteria, such as decisional conflict or stage of decision making. This shows that despite its buffering capacity on psychological burden, possessing coping self-efficacy does not lead to more decisiveness in preference-sensitive decisions.

## 1. Introduction

*BRCA1* and *BRCA2* are tumor suppressor genes in humans. A pathogenic variant (PV) in either one of these genes leads to impaired DNA repair and, subsequently, an increased risk of lifetime breast and ovarian cancer [1]. Female *BRCA1/2* PV carriers face a lifetime risk of roughly 70% for breast cancer (*BRCA1*: 95% confidence interval (CI) 65–79; *BRCA2* 95% CI 61–77). For ovarian cancer, women with *BRCA1* PV have a cumulative lifetime risk of 44 % (95% CI 36–53), whereas *BRCA2* PV carriers carry a risk of 17% (95% CI 11–25) [2]. By contrast, women in the general population carry a lifetime breast cancer risk of roughly 10–15 % and a lifetime ovarian cancer risk of just 1%. *BRCA1* and *BRCA2* run in families, with a 50% chance of passing the PV to offspring [1,2]. Women who are subsequently found to carry a *BRCA1/2* PV engage in complex preventive decision making to address their individual cancer risks [3]. They may opt for risk-reducing surgeries, such as bilateral mastectomy or salpingo-oophorectomy, or intensified breast surveillance (IBS). Additionally, they may be offered chemoprevention, such as a daily dosage of tamoxifen or raloxifene, which both interfere with estrogen in tumor genesis [4]. 

All of these options have different benefit–harm profiles [5,6]. For the prevention of breast cancer, risk-reducing bilateral mastectomy is an effective procedure to reduce breast cancer incidence for both *BRCA1* and *BRCA2* PV carriers and breast cancer mortality for *BRCA1* PV carriers [7,8]. However, the removal of healthy breast tissue may have adverse effects on the ability to breastfeed, body image, and sexual satisfaction [9,10]. On the other hand, IBS is considerably less invasive and effectively identifies breast cancer in its early stages about 80% of the time [11]. Nevertheless, there is a remaining risk of finding breast cancer in need of aggressive treatment or even too late for effective treatment. Some women may opt for IBS to postpone their ultimate decision for risk-reducing mastectomy [12]. For the prevention of ovarian cancer, there is no effective surveillance program [13,14]. Therefore, the only preventive procedure currently available to reduce ovarian cancer incidence and mortality is salpingo-oophorectomy, i.e., the removal of the ovaries and fallopian tubes. While this surgical procedure has been shown to reduce ovarian cancer incidence and mortality [15], it comes with the downside of permanently ending the ability to bear children as well as the induction of surgical menopause in otherwise premenopausal women [16,17]. The abovementioned options may also induce adverse physical and emotional consequences. As such, these trade-offs need to be considered by *BRCA1/2* PV carriers and display the complexity of preventive decision making. Additionally, women from high-risk families have usually experienced cancer cases in their female family members. Due to this, undergoing testing for a *BRCA1/2* PV and finding out about a confirmed PV status may induce a variety of emotions in cancer-unaffected women, including anxiety, cancer worry, and even depressive symptoms [18]. Few women even go as far as describing the process as traumatic [19]. *BRCA1/2* PV carriers have also been reported to suffer from sleeping problems or poor sleep quality [20]. Feelings of uncertainty associated with the advantages and disadvantages of each preventive option that each woman weighs differently based on her individual values, also known as decisional conflict, may arise [21,22]. Consequently, high levels of decisional conflict may result in decision delay or failure of decision implementation. 

Self-efficacy is generally defined as the capability to believe in oneself to execute behaviors that lead to one’s goals [23]. In women with breast cancer, an intervention that addressed self-efficacy led to a higher quality of life and less symptom distress [24]. This is indicative of a relationship between self-efficacy and psychological outcomes. Self-efficacy has also previously been studied in the context of genetic testing uptake among individuals of Ashkenazi Jewish descent, who are considered at high risk for carrying a *BRCA1/2* PV. It was found that higher self-efficacy was associated with higher intention to undergo genetic testing [25]. When investigating women with a confirmed *BRCA1/2* PV in Norway, women with higher self-efficacy showed less anxiety regardless of cancer history [26]. However, the women in this sample received their genetic test result on average 5 years prior to participating in the study and are therefore not representative of women with a newly found PV. Coping self-efficacy may be understood as a subconcept of general self-efficacy and denotes one’s belief in the ability to cope effectively with a challenge or unexpected event, such as a medical diagnosis [27]. In light of the fact that receiving a positive genetic test result can induce significant distress, coping self-efficacy might serve as a buffer to some of this burden.

To the best of our knowledge, there are no studies investigating coping self-efficacy after receiving a genetic test result and its potential to buffer the psychological impact of a positive genetic test result. Therefore, the aim of the current study is to investigate the relationship between coping self-efficacy and psychological morbidity in cancer-unaffected *BRCA1/2* PV carriers. Based on previous findings regarding the relationship between self-efficacy and psychological outcomes, the following is predicted: among *BRCA1/2* PV carriers, higher coping self-efficacy is related to (1) less anxiety; (2) less depression; (3) less decisional conflict; (4) less impact of genetic test result disclosure; and (5) being further along the decision-making process compared to *BRCA1/2* PV carriers with lower self-efficacy.

## 2. Materials and Methods

### 2.1. Setting and Eligibility Criteria

The study was conducted between 2019 and 2021 at six study centers in Germany that belong to the Familial Breast and Ovarian Cancer consortium (University Hospital of Cologne, University Hospital of Würzburg, University Hospital Schleswig-Holstein Campus Kiel, University Hospital Heidelberg, University Hospital Carl Gus-tav Carus Dresden, and University Hospital rechts der Isar Munich). Data presented in this paper were collected as part of a larger randomized controlled trial evaluating a decision support program in healthy *BRCA1/2*-positive women (EDCP-BRCA) [28]. This secondary analysis will focus on subgroup data from women with a newly found *BRCA1/2* PV. The inclusion criteria for this subset were (1) newly found *BRCA1/2* PV at maximum 6 weeks prior; (2) age between 25 and 60 years; (3) no prior history of breast or ovarian cancer; and (4) sufficient knowledge of the German language. Women were excluded if (1) they had a history of breast and/or ovarian cancer; (2) had an unclear sequence variant in the *BRCA1/2* genes; (3) had insufficient knowledge of the German language; (4) had cognitive impairments; or (5) had debilitating diseases or cancers. Women were recruited after receiving their genetic test result after written informed consent was obtained.

### 2.2. Data Collection and Measures

The data were collected from a self-administered questionnaire that participants received after giving informed consent. Participants were able to take the questionnaire home and were asked to return the questionnaire within one week. Data were collected on age, marital status, highest level of education, employment status, and children. Additionally, several measures on self-efficacy, decision making, and psychological burden were included: Coping self-efficacy. Self-efficacy to cope with the genetic test result was measured using the Coping Self-Efficacy Scale (CSE) [29]. The 13-item CSES consists of three subscales: problem-focused coping (6 items); stopping unpleasant emotions and thoughts (4 items); and support from friends and family (3 items). Higher scores indicate feeling better equipped to handle the genetic test result, whereas lower scores indicate worry that one might not be able to cope with the test result well.Anxiety and depression. The German version of the Hospital Anxiety and Depression Scale (HADS) was used to assess the level of psychological distress following disclosure of the genetic test result [30,31]. The HADS is comprised of 14 items that form two scales measuring levels of depression and anxiety, respectively. For both scales, a sum score of 0 to 7 for either subscale is considered nonelevated by the authors, sum scores between 8 and 10 are considered conspicuous, suggesting borderline clinically relevant levels of depression and anxiety, while a score ≥11 indicates clinically relevant depression or anxiety on their respective scale.Impact of genetic test result. To measure the subjective distress of the genetic test result, the Impact of Event Scale (IES) was used [32,33]. The 22-item scale is used to evaluate the impact of potentially traumatic events and consists of three subscales: intrusion, avoidance, and hyperarousal. The total score can range from 0 to 88 with high scores indicating symptoms of post-traumatic stress. A cut-off score of 33 is usually recommended for a case of post-traumatic stress. An equation to detect those with symptoms severe enough to fulfill criteria for a suspected diagnosis of post-traumatic stress disorder (PTSD) from the three subscales has been published and was utilized [34].Status of decision. The Stage of Decision-Making Scale (SDMS) is a single-item scale that assessed how far along women were in their decision-making process on how to deal with their cancer risk [35]. The scale has four response options ranging from “I haven’t begun to think about the choices” to “I have already made my choice”.Decisional conflict. Decisional conflict captures the uncertainty in health- or treatment-related decisions. It was measured using the German translation of the Decisional Conflict Scale (DCS) [36,37]. The scale comprises five subscales: feeling uninformed, uncertainty about the best choice, lack of support, unclarity about personal values, and effective decision. Subscales are summarized to yield an overall score. The total score can range from 0 to 100 with higher values indicating higher decisional conflict. The authors suggest that a total score under 25 indicates no decisional conflict and is associated with decision implementation, whereas scores ≥ 37.5 indicate insecurity about decision implementation and may result in decision delay.

### 2.3. Statistical Methods

All statistical analyses were performed using SPSS Version 27.0 ) (IBM, Armonk, New York, USA). All results were interpreted using a two-sided *p*-value < 0.05. Established cut-off scores for anxiety, depression, and post-traumatic stress were used, and categories were built accordingly. Separate univariate ANOVA (in case of three or more groups) or *t*-tests (in case of two groups) with these categories used as independent variables and coping efficacy as a dependent variable were performed. Bonferroni-corrected alpha levels were calculated and reported in cases of post hoc contrasts and multiple comparisons.

## 3. Results

Overall, 130 women with a newly found *BRCA1/2* PV and no personal history of cancer were enrolled. The average time between genetic test result disclosure and returning the questionnaire was M = 22 days (SD = 19 days). Women’s ages ranged from 24 to 60 years (M = 36 years; SD = 9.37 years). Further demographic data collected are presented in Table 1, and a correlation matrix for all psychological measures with means and standard deviations is presented in Table 2.

### 3.1. Demographic Data

To identify potential covariates in the relationship between coping self-efficacy and psychological morbidity, age, BRCA mutation, and parity were analyzed. Age was not significantly correlated to any variable of interest. When comparing *BRCA1* and *BRCA2* pathogenic variant carriers, no differences in terms of any psychological outcomes were found. When comparing women with to women without children, no differences were found. Therefore, no potential covariates were identified. 

### 3.2. Anxiety and Coping Self-Efficacy

The sample’s overall anxiety mean measured by the HADS was just under the threshold of borderline clinical anxiety at *M* = 7.81 (SD = −3.55). Anxiety was significantly inversely correlated with problem-based coping, r(130) = −0.37, *p* < 0.001; support from family and friends, *r*(130) = −0.31, *p* < 0.001; and stopping negative emotions, *r*(130) = −0.52, *p* < 0.001 measured by the CSE. 

When categorized into groups, almost half of the sample (45.4%, *n* = 59) did not show elevated anxiety (score ≤ 7). Almost a third (32.3%) of participants showed conspicuous, borderline clinical anxiety (score ≥ 8 ≤ 10), and 22.3% of the sample experienced a clinical level of anxiety (score ≥ 11). Means for all groups per subscale and overall scale are displayed in Table 3. Four separate univariate ANOVAs with the three anxiety categories (non-elevated, borderline, and clinical anxiety) used as factors and the subscales as well as the overall scale of the CSE used as the dependent variable were performed. All ANOVAs revealed significant differences between the three anxiety categories: problem-based coping (CSE-PF), *F*(129) = 7.97, *p* < 0.001, η² = 0.11; support from family and friends (CSE-SFF), *F*(129) = 7.66, *p* < 0.001, η² = 0.11; stopping unpleasant emotions (CSE-SUE), F(129) = 15.92, *p* < 0.001, η² = 0.20; and overall coping self-efficacy (CSE-T), *F*(128) = 16.88, *p* < 0.001, η² = 0.21. For the following post hoc tests, a Bonferroni-adjusted α = 0.016 (0.05/3) was calculated based on the three comparisons per dependent variable (non-elevated vs. borderline clinical, borderline clinical vs. clinical, and non-elevated vs. clinical).

For problem-based coping (CSE-PF), post hoc tests showed that participants with non-elevated anxiety were significantly better at problem-focused coping when compared to participants with clinical anxiety, *M_diff_* = 8.06, *SD_diff_* = 2.0, *p* < 0.001. There was no significant difference for non-elevated vs. borderline clinical anxiety or borderline clinical vs. clinical anxiety. 

For support from family and friends (CSE-SFF), post hoc tests revealed that those with non-elevated anxiety showed higher confidence in receiving support from friends and family compared to those with clinical anxiety, *M_diff_* = 4.59, *SD_diff_* = 1.23, *p* < 0.001. There was also a difference between participants with non-elevated anxiety vs. participants with borderline clinical anxiety, *M_diff_* = 2.72, *SD_diff_* = 1.10, *p* = 0.014, but no difference between participants with borderline clinical vs. clinical anxiety.

For stopping unpleasant emotions (CSE-SUE), post hoc tests revealed that those with non-elevated anxiety felt more equipped in stopping unpleasant emotions compared to those with clinical anxiety, *M_diff_* = 9.30, *SD_diff_* = 1.77, *p* < 0.001. Moreover, participants with non-elevated anxiety also felt more equipped to stop negative emotions than participants with borderline clinical anxiety *M_diff_* = 6.03, *SD_diff_* = 1.57, *p* < 0.001. Like the other two subscales, there was no significant difference between participants with borderline clinical and clinical anxiety. 

Overall coping self-efficacy was better for non-elevated vs. borderline clinical anxiety, *M_diff_* = 12.80, *SD_diff_* = 3.59, *p* = 0.002, and for non-elevated vs. clinical anxiety, *M_diff_* = 22.52, *SD_diff_* = 4.03, *p* < 0.001. The difference between borderline clinical and clinical anxiety was not significant. The effect sizes from all ANOVAs ranged from η² = 0.11 to η² = 0.21, indicating a medium to large effect.

### 3.3. Depression and Coping Self-Efficacy

The sample showed a mean score of depression of *M* = 3.40, SD = 3.08. Depression was significantly inversely correlated with problem-based coping, r(130) = −0.45, *p* < 0.001; support from family and friends, *r*(130) = −0.52, *p* < 0.001; and stopping negative emotions, *r*(130) = −0.46, *p* < 0.001. The prevalence of clinical depression (3.1%) and borderline clinical depression (9.6%) was low in this sample. Due to only 13 participants fulfilling the criteria for borderline clinical and clinical depression, the two categories were combined into one. Upon inspection of the data, variances within the two depression categories were determined to be equal. Therefore, *t*-tests with a Bonferroni-adjusted alpha level of α = 0.0125 (0.05/4) were used to determine the differences in the subscales of coping self-efficacy. When compared to participants with non-elevated depression scores, participants with elevated depression scores showed less problem-based coping, t(127) = 5.21, *p* ≤ 0.001, had less confidence in their social support, *t*(128) = 5.75, *p* ≤ 0.001, and felt less equipped in stopping negative emotions, *t*(128) = 3.53, *p* ≤ 0.001. Overall, participants with elevated depression exhibited less coping self-efficacy, *t*(128) = 5.93, *p* ≤ 0.001, compared to participants with non-elevated depression scores. Means are presented in Table 4.

### 3.4. Impact of Event Scale

The mean score for the impact of event scale was *M* = 11.57 (SD = 6.49) for intrusion, *M* = 12.01 (SD = 7.34) for avoidance, and *M* = 8.54 (SD = 7.39) for hyperarousal. The combined mean score for the whole sample was *M* = 32.25, *SD* = 17.24. When categorizing participants according to the threshold of meaningful post-traumatic stress (score of ≥33), 41.5% of participants showed post-traumatic stress after genetic test result disclosure, whilst 58.5% did not. Coping self-efficacy was inversely correlated with the impact of the event, suggesting that those with higher self-efficacy experienced less post-traumatic stress, *r*(127) = −0.48, *p* < 0.001. Multiple comparisons revealed that subscale scores were lower in participants who experienced a meaningful case of post-traumatic stress compared to those that did not: support from family and friends, *M_diff_* = 3.56, *SD_diff_* = 0.97, *p* = 0.005; stopping negative emotions, *M_diff_* = 6.08, *SD_diff_* = 1.45, *p* = 0.005. The difference for problem-focused coping did not reach significance, *M_diff_* = 3.36, *SD_diff_* = 1.66, *p* = 0.12. The equation to detect those with clinical cases of PTSD was calculated for each participant [34]. Nine participants (6.9%) showed symptoms severe enough to fulfill criteria for a suspected diagnosis of PTSD according to Maercker and Schützwohl [34]. 

### 3.5. Decision Making and Coping Self-Efficacy

Frequency analysis of the SDMS (missing n = 7) yielded that very few participants (3.8%) had not thought about the preventive options yet (stage 1). Most participants (42.3%) reported that they were currently thinking about the different preventive options (stage 2), while 16.9% reported being close to making a decision (stage 3). About a third of participants (31.9%) indicated that they had already made a choice about which preventive option they were going to opt for (stage 4). While we hypothesized that those with higher coping self-efficacy would be further along in the decision-making process, univariate ANOVA yielded no significant differences between the groups regarding any of the coping self-efficacy subscales or the total scale.

In terms of decisional conflict, over half of the sample (51.5%) had a total score ≥37.5, indicating the presence of decisional conflict. When investigating the subscales, 36.9% felt uninformed, 49.2% felt unclear about personal values, 46.2% felt unsupported in the decision-making process, and 60% felt uncertain about the best preventive option. As many participants had not made a final choice about the best preventive option, the subscale effective decision was not analyzed. Total decisional conflict was unrelated to coping self-efficacy, *r*(128) = −0.12, *p* = 0.17, and neither of the subscales showed any significant correlation (see Table 2).

## 4. Discussion

In this study, we identified high rates of anxiety, with over half of newly found cancer-unaffected *BRCA1/2* PV carriers showing borderline clinical or clinical anxiety. However, increased rates of clinical anxiety have previously been reported for newly found *BRCA1/2* PV carriers and are most likely associated with cancer worry in this population [38,39,40].

The present study addresses a gap in the literature by investigating coping self-efficacy and its relationship with psychological and decision-related outcomes after genetic test result disclosure in cancer-unaffected female *BRCA1/2* PV carriers. Overall, those with higher coping self-efficacy were identified to be more robust in processing a positive genetic test result but were not more decisive. Additionally, higher coping self-efficacy was not related to more certainty about preventive decision making compared to lower coping self-efficacy. In line with the study’s hypotheses, coping self-efficacy was negatively correlated with the experience of psychological distress in newly *BRCA1/2*-positive women. Those with higher coping self-efficacy, especially those with higher confidence in stopping negative emotions and thoughts, felt less anxious. Moreover, those with confidence to receive social support and those that feel equipped in their problem-solving skills experienced less anxiety. 

For depression, a similar pattern of results emerged, with lower coping self-efficacy being associated with a higher likelihood of clinical or borderline clinical depression. Participants with high confidence to receive social support from family and friends post genetic test result felt significantly less depressed. This pattern has similarly been reported in the literature; breast cancer patients with higher coping self-efficacy generally experience less psychological distress, including depression, anxiety, distress, and cancer worry [41,42]. Our study expands on this literature by showing that not only breast cancer patients but also healthy women at high risk for breast cancer experience less distress with higher coping self-efficacy, demonstrating that coping self-efficacy is not only effective in cases of an actual diagnosis. These results are consistent with the general self-efficacy theory, which asserts that efficacious people experience less distress and adjust better in stressful situations [23].

While coping self-efficacy seemed to have a strong negative relationship to psychological burden after genetic test result disclosure, it did not appear to have any impact on the current certainty in the decision-making process. Decisional conflict was present for a majority of the sample, but contrary to the hypotheses, higher coping self-efficacy was not associated with less decisional conflict, nor was it associated with being further along the decision-making process. Interestingly, decisional conflict was by and large not related to any of the psychological parameters. Psychological distress has previously been found to be a predictor of decisional conflict in longitudinal studies [43,44]. A possible reason for the discrepancy between those findings and the results from this present study could be that these studies investigated cancer patients or cancer survivors and not healthy high-risk individuals. Another reason could be the cross-sectional analysis of baseline data, which limits the informative value of this relationship.

Further limitations of this study need to be addressed. Participants were recruited after receiving their genetic test result. This could have led to biased sampling because women with seemingly higher psychological distress might not have been asked to participate in the study, as well as self-selection because distraught women might not have been willing to participate. Additionally, it has previously been reported that those with higher self-efficacy are more likely to undergo genetic testing for *BRCA1/2* in the first place [25], potentially skewing the sample toward women with higher self-efficacy. The sample was highly educated, with almost half of the participants possessing a university degree, and, therefore, might not be entirely representative of the general population of cancer-unaffected *BRCA1/2* PV carriers. As mentioned above, this secondary analysis only investigated baseline data. Due to this, we are unable to determine if coping self-efficacy would be a protective factor for psychological morbidity in the long run. Additionally, in order to assess the full effect on preventive decision making, a follow-up period would be desirable. Future studies could investigate how higher coping self-efficacy influences the preventive decision-making process and how it affects psychological outcomes over a longer period.

Despite these limitations, several clinical implications may be deducted from our results. The high rates of clinical anxiety support the notion for low-threshold psycho-oncological support, which should routinely be offered to newly diagnosed *BRCA1/2* PV carriers. While this is already available in some countries, a broad rollout would be desirable. In the design of such psycho-oncological support, self-efficacy intervention components may be included to alleviate some of the psychological morbidity that newly found carriers experience. Decision-support programs, such as the use of decision aids [22,45,46], or decision coaching [28] for this population may equally address psychological burden as well as decision uncertainties.

## 5. Conclusions

The aim of this study was to obtain insights into the relationship between coping self-efficacy and psychological morbidity in newly found *BRCA1/2* PV carriers, such as clinical anxiety, depression, and post-traumatic stress. Additionally, we investigated whether coping self-efficacy would be related to decision making between the different available preventive options in the sense that individuals with higher coping self-efficacy experience less decisional conflict and are closer to making a final decision. Based on the results of the present study, we conclude that women experience high levels of psychological morbidity after receiving a positive *BRCA1/2* test result. However, the likelihood of experiencing high levels of anxiety, depression, and post-traumatic stress was lower in individuals with higher coping self-efficacy. By contrast, coping self-efficacy was not associated with any decision-related outcomes. It was not related to less decisional conflict and was not associated with being closer to making a preventive decision.

## Figures and Tables

**Table 1 ijerph-20-01684-t001:** Demographic frequencies in the sample.

Total	n	%
	130	100
**Education**		
Secondary education *(9 or 10 years)*	32	24.6
High school degree/A-Levels *Abitur*	36	27.7
University degree	62	47.7
**Marital status (n = 2 missing)**		
Single	48	36.9
Married or in a long-term relationship	72	55.4
Divorced	8	6.2
**Parity (n = 2 missing)**		
No children	64	49.2
1 child	26	20.0
2 children	30	23.1
3+ children	8	6.2
**Family planning completed (n = 15 missing)**		
Yes	57	43.8
No	58	44.6

Note. No demographic variable showed significant differences within its subgroups in terms of coping self-efficacy.

**Table 2 ijerph-20-01684-t002:** Means, standard deviation, and correlations among all continuous psychological and decision-making variables.

Variables	*M*	*SD*	1	2	3	4	5	6	7	8	9	10	11	12	13	14	15
HADS—Depression	3.40	3.08	1														
2. HADS—Anxiety	7.81	3.55	0.70 **	1													
3. DCS—Informed	31.60	17.23	0.04	0.03	1												
4. DCS—Value Clarity	38.04	22.71	0.05	0.18 *	0.56 **	1											
5. DCS—Support	36.95	19.08	0.11	0.20 *	0.53 **	0.63 **	1										
6. DCS—Uncertainty	46.29	23.89	0.08	0.23 **	0.54 **	0.77 **	0.71 **	1									
7. DCS—Total	37.87	17.95	0.07	0.19 *	0.73 **	0.88 **	0.84 **	0.91 **	1								
8. CSE—PF	41.28	9.45	−0.45 **	−0.37 **	−0.14	−0.17	−0.06	−0.09	−0.13	1							
9. CSE—SFF	22.65	5.71	−0.52 **	−0.31 **	−0.05	0.01	−0.03	0.10	0.02	0.59 **	1						
10. CSE—SUE	20.32	8.65	−0.46 **	−0.52 **	−0.07	−0.15	−0.18 *	−0.13	−0.15	0.58 **	0.40 **	1					
11. CSE—Total	84.43	19.80	−0.57 **	−0.51 **	−0.11	−0.14	−0.13	−0.07	−0.12	0.90 **	0.74 **	0.82 **	1				
12. IES—Intrusion	11.57	6.48	0.40 **	0.53 **	−0.03	0.01	0.09	0.07	0.04	−0.19 *	−0.19 *	−0.33 **	−0.29 **	1			
13. IES—Avoidance	12.09	7.34	0.36 *	0.42 **	0.09	0.06	0.18 *	0.07	0.12	−0.31 **	−0.30 **	−0.33 **	−0.38 **	0.38 **	1		
14. IES—Hyperarousal	8.54	7.39	0.62 **	0.69 **	0.03	0.07	0.10	0.13	0.09	−0.35 **	−0.35 **	−0.45 **	−0.47 **	0.66 **	0.46 **	1	
15. IES—Total	31.95	17.31	0.57 **	0.66 **	0.04	0.06	0.17 *	0.12	0.11	−0.35 **	−0.34 **	−0.47 **	−0.47 **	0.80 **	0.76 **	0.87 **	1

Note. HADS = Hospital Anxiety and Depression Scale [31]. DCS—Decisional Conflict Scale [36]. CSE—Coping Self-Efficacy Scale [29]. PF—Problem-focused coping subscale of the CSE. SFF—Support from family and friends subscale from the CSE. SUE—Stopping unpleasant emotions and thoughts subscale of the CSE. IES—Impact of Event Scale [32]. * *p* ≤ 0.05; ** *p* ≤ 0.01.

**Table 3 ijerph-20-01684-t003:** Mean scores on the CSE according to anxiety level.

	Groups					
	Non-Elevated Anxiety HADS-A Score ≤ 7 n = 59		Borderline Clinical Anxiety HADS-A Score ≥ 8 ≤ 10 n = 42		Clinical Anxiety HADS-A Score ≥ 11 n = 29	
	*M*	*SD*	*M*	*SD*	*M*	*SD*
CSE-PF	44.20	8.98	40.71	8.72	36.14	9.33
CSE-SFF	24.56	4.18	21.83	6.04	19.97	6.65
CSE-SUE	24.34	8.44	18.31	6.58	15.03	8.01
CSE-T	93.66	16.13	80.86	16.85	71.14	21.62

Note. HADS-A = Anxiety subscale of the Hospital Anxiety and Depression Scale [31]. CSE-PF = Problem-focused coping subscale of the CSE [29]; CSE-SFF = Support from family and friends subscale from the CSE; CSE-SUE = Stopping unpleasant emotions and thoughts subscale of the CSE; CSE-T = Total score of the CSE.

**Table 4 ijerph-20-01684-t004:** Mean scores on the CSE according to depression level.

	Groups			
	Nonelevated Depression HADS-D Score ≤ 7 n = 117		Elevated Depression HADS-D Score ≥ 8 n = 13	
	*M*	*SD*	*M*	*SD*
CSE-PF	42.59	8.70	29.46	7.78
CSE-SFF	23.51	4.98	14.92	6.22
CSE-SUE	21.17	8.47	12.62	6.21
CSE-T	87.50	17.63	57.00	17.21

Note. HADS-D = Depression subscale of the Hospital Anxiety and Depression Scale [31]. CSE-PF = Problem-focused coping subscale of the CSE [29]; CSE-SFF = Support from family and friends subscale from the CSE; CSE-SUE = Stopping unpleasant emotions and thoughts subscale of the CSE; CSE-T = Total score of the CSE.

## Data Availability

All data are available from authors upon reasonable request.

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
