# Peer review of "Coping Self-Efficacy and Its Relationship with Psychological Morbidity after Genetic Test Result Disclosure: Results from Cancer-Unaffected BRCA1/2 Mutation Carriers"

_ijerph, 2023, doi:10.3390/ijerph20031684_

Round 1

Reviewer 1 Report

The topic is of major interest and the authors do a very good job explaining the rationale for the study. However, the authors did not make the optimal choice of data analysis procedures. Running multiple, separate group comparisons is problematic because it leads to an increased probability of Type I error. Additionally, each relationship is considered separately, therefore the study does not inform on the strength of the relationships when variables are taken into account simultaneously. 

Results from the group comparisons are correctly reported but I suggest including some additional analyses such as multiple regression, path analysis or structural equation modeling to simultaneously assess the relationships between variables and identify the strongest predictors. 

Author Response

Dear reviewer,

Thank you very much for your thoughtful review of our manuscript. We agree that multiple comparisons are problematic, which is why every p-value is Bonferroni-adjusted to correct for type one error inflation. Bonferroni-correction is widely considered as the most conservative correction; therefore, we trust that this correction lead to in fact significant results. The results we found are consistent with psychological literature, we therefore do not think that a type one error occurred.

We appreciate your suggestion on running multiple regression, path analysis or structural equation modeling. However, after receiving statistical consultation, we think it would be problematic to try and identify predictors when data on all variables were gathered simultaneously. The data is in fact cross-sectional and therefore not ideal to establish causal relationships between variables. We do however keep your idea in mind for when we publish longitudinal data of our study.

We again would like to thank you for your review and wish you all the best!

The authors

Reviewer 2 Report

The premise of this study could potentially contribute some interesting information to our understanding of how genetic test (BRCA1/2 PV) and higher coping self-efficacy influence psychological outcome and preventive decision-making. Previous research had been well-documented that patient with BRCA1/2 carriers  increase in mental distress but did not impact their long-term psychologic outcome and clinically dysfunction. The current consensus is seemed similar to your study.  The association between Coping Self-Efficacy and psychological outcome was discussed in different medical disease. Major points that need to be addressed include the following:

1.      Why genetic test and higher coping self-efficacy did not affect clinically decision-making ?

2.      What is the clinical application of the results of this research?

Author Response

Dear reviewer,

we thank you for your thoughtful review of our manuscript. We agree that the current consensus in the literature is similar to the results of our study. In the following, we would like to address your questions:

Why genetic test and higher coping self-efficacy did not affect clinically decision-making?

Unfortunately, we do not have a good answer for this (yet). There are multiple possible explanations for this phenomenon. First, it could be possible that psychological well-being is less related to decision-making than previously thought. Second, it seems that the studies that previously found a relationship between these two variables are conducted in cancer patients and not healthy mutation carriers. It seems possible that there are distinct differences between these populations that explain this discrepancy. We mention this as a possible explanation in lines 340-341.

What is the clinical application of the results of this research?

We mention in lines 362-364 that the psychological morbidity in these women is relatively high and therefore recommend psychological support as part of routine care. Furthermore, in lines 365-367 we suggest that self-efficacy interventions may be a low-threshold way to improve psychological morbidity in this population.

We again thank you for your review and wish you all the best!

The authors

Reviewer 3 Report

Dear authors,

Thank you for the opportunity to revise your manuscript. I appreciated reading your manuscript, which aims to longitudinally assess the coping self-efficacy, anxiety, depression and the status of the decision, and the decisional conflict after the diagnosis of a BRCA1/2 mutation in a cohort of 130 German women.

I listed my concerns about your manuscript.

1.       In my opinion, the study's aim is unclear since you espose your hypothesis rather than the scope of the manuscript.

2.       In line 71, I would also suggest that BRCA1/2 women suffer from sleep problems or poor sleep quality: https://doi.org/10.1038/s41598-022-25014-7,  Especially the first offers a solution for improving sleep problems.

3.       The statistical analysis should be better described.

4.       How many days passed between the diagnosis and the compilation of the questionnaire? The compilation latency could have influenced all the analysed variables because those diagnosed early could have familiarised with the mutations and be nearer to a decision.

5.       Did you reflect on

a.       performing a chi-squared analysis between all the categories?

b.       Using the age as a covariate?

c.       Creating age categories and comparing them? Younger women could have more trouble with the decision than the oldest ones, because the latter could already have given birth to children and have a family, and the surgical option could be more accessible. Younger women, probably not having children or a family, need to reflect more on the surgical option.

d.    analysing the BRCA1 separately form the BRCA2 since the two mutations could have different consequences, tumour risks, prognosis, and counselling?

6.       At the beginning of the discussion section, you wrote about the comparisons between your sample and the general German population. However, this analysis is not reported in the results section.

7.       I suggest you find more and better applicative suggestions to your manuscript and results. As the last lines of the discussion section appear, psychological support for these women has already been advanced and plied.

Author Response

Dear reviewer,

Thank you very much for your thoughtful review of our manuscript. It really helped us a lot in improving it. We firstly would like to establish that this is by no means a longitudinal study, as your reviewer report suggests. We do not want any misunderstandings about this, our study is purely cross-sectional. In the following, we will be addressing your concerns:

In my opinion, the study's aim is unclear since you espose your hypothesis rather than the scope of the manuscript.

We happily included a sentence that describes the study’s aim more thoroughly.

In line 71, I would also suggest that BRCA1/2 women suffer from sleep problems or poor sleep quality: https://doi.org/10.1038/s41598-022-25014-7,  Especially the first offers a solution for improving sleep problems.

Thank you for this great paper, which we happily included.

The statistical analysis should be better described. How many days passed between the diagnosis and the compilation of the questionnaire? The compilation latency could have influenced all the analysed variables because those diagnosed early could have familiarised with the mutations and be nearer to a decision.

This was indeed mentioned in the beginning of the result section (lines 170-171): Average time between genetic test result disclosure and returning the questionnaire was M = 22 days (SD = 19 days). We purposefully did only include those women with a new test result and did not include women who received their test result more than 6 weeks ago.

Did you reflect on

  1. performing a chi-squared analysis between all the categories?
  2. Using the age as a covariate?
  3. Creating age categories and comparing them? Younger women could have more trouble with the decision than the oldest ones, because the latter could already have given birth to children and have a family, and the surgical option could be more accessible. Younger women, probably not having children or a family, need to reflect more on the surgical option.
  4. analysing the BRCA1 separately form the BRCA2 since the two mutations could have different consequences, tumour risks, prognosis, and counselling?

Thank you for these great suggestions. We did reflect on using chi-square analyses. Unfortunately, these are only applicable if relationship between two categorical variables needs to be assessed, which was not the case for our study. For example, no meaningful cut-off scores exist for coping self-efficacy, therefore no groups were formed. We could have used chi-square to look at the relationship of anxiety and depression. This is however, not within the scope of the current manuscript and would not provide additional value. In terms of your points (b) and (c), we looked at age and found that it was not significantly correlated to any of the outcomes and therefore did not include it as a factor. Furthermore, we also checked if there was a difference between carriers of genes BRCA1 and BRCA2 but found no differences in terms of (1) anxiety, (2) depression, (3) decisional conflict, (4) coping self-efficacy or (5) impact of event. We will happily add some sentences in the beginning of the result section explaining that age and BRCA mutation did not make a difference (see new manuscript section 3.1.)

At the beginning of the discussion section, you wrote about the comparisons between your sample and the general German population. However, this analysis is not reported in the results section.

Thank you for bringing this up. This comparison is not based on an analysis in our paper. We see how the placement in the first section of the discussion may be confusing. We therefore removed the comparison to the German population.

I suggest you find more and better applicative suggestions to your manuscript and results. As the last lines of the discussion section appear, psychological support for these women has already been advanced and plied.

Thank you for this suggestion. Unfortunately, we do not think that this can be stated for all countries and healthcare systems. We added a sentence that it may be available to all in some countries. Additionally, we do think that self-efficacy interventions would a novelty for this population as a whole.

We wish to thank you again for reviewing our manuscript and wish you all the best!

The authors

Round 2

Reviewer 1 Report

No significant changes were made to the manuscript; therefore, my recommendation remains the same.

Author Response

Dear reviewer, 

we would to once again thank you for your time in reviewing our manuscript. We would like to address your comments on the first draft, as your second review did not take our explanations provided into consideration.

The topic is of major interest and the authors do a very good job explaining the rationale for the study.

Thank you.

However, the authors did not make the optimal choice of data analysis procedures. Running multiple, separate group comparisons is problematic because it leads to an increased probability of Type I error.

We agree that multiple comparisons are problematic, which is why every p-value is Bonferroni-adjusted to correct for type one error inflation. Bonferroni-correction is widely considered as the most conservative correction; therefore, we trust that this correction led to in fact significant results. The results we found are consistent with psychological literature, we therefore conclude that it is unlikely that these results are due to a type one error.

Results from the group comparisons are correctly reported but I suggest including some additional analyses such as multiple regression, path analysis or structural equation modeling to simultaneously assess the relationships between variables and identify the strongest predictors. 

We would like to understand which variables you would like us to investigate with these additional analyses. You mention that you would like to see analyses “to simultaneously assess the relationships between variables and identify the strongest predictors” – predictors of which variable exactly? Coping self-efficacy was not conceptualized as the outcome variable, but instead as the predictor variable itself. The outcomes were various psychological morbidities, such as anxiety, depression, or cancer distress. We therefore do not have various predictor variables that we could reasonably enter into a regression model. Additionally, we do not think that these analyses would be better suited to answer our prospectively defined hypotheses and changing or editing hypotheses to match post hoc analyses infringes upon scientific integrity.

Kindly, we ask you to provide a more detailed explanation on which, if any, additional analyses you suggest after taking our rationale into account.

Nevertheless, we thank you again for reviewing our manuscript and wish you all the best!

Kind regards

The authors

Reviewer 2 Report

I appreciate the great efforts that the authors have made in response to my questions and concerns. The revision clarifies almost all the points I raised. I have no further question.

Author Response

Dear reviewer, 

thank you very much for your kind words and your thoughtful inputs on our manuscript. 

All the best

The authors

Reviewer 3 Report

Dear authors,

The manuscript significantly improved after the revision.

Before final acceptance for publication, I would like you to express the recruitment time more clearly.

In the manuscript, you reported this inclusion criterion: "(1) newly found BRCA1/2 PV"; however, in the author's reply, you wrote this sentence after my request about the time between the diagnosis and the recruitment: "We purposefully did only include those women with a new test result and did not include women who received their test result more than 6 weeks ago."

If 6 weeks is the minimum time after the diagnosis to be recruited, could you please insert this sentence in the manuscript?

Author Response

Dear reviewer, 

thank you for your thoughtful input. 

We have added a sentence about the 6 week maximum to the manuscript (line 111).

All the best

The authors